# Clinical Usefulness of the Valsalva Manoeuvre to Improve Hemostasis during Thyroidectomy

**DOI:** 10.3390/jcm11195791

**Published:** 2022-09-29

**Authors:** Mario Pacilli, Giovanna Pavone, Alberto Gerundo, Alberto Fersini, Antonio Ambrosi, Nicola Tartaglia

**Affiliations:** Department of Medical and Surgical Sciences, University of Foggia, Luigi Pinto Street, No. 1, 71122 Foggia, Italy

**Keywords:** Valsalva manoeuvre, hemostasis, surgical drains, thyroidectomy, bleeding

## Abstract

Bleeding after total thyroidectomy remains a rare event that affects early postoperative morbidity, occurring in 0.3% to 4.2% of cases. Intraoperative bleeding is an unpleasant complication, and it is often easily manageable, although postoperative bleeding may represent a life-threatening condition for the patient. The purpose of our study was to clarify the role of the Valsalva manoeuvre to reduce postoperative bleeding. Between January 2019 to February 2022, 250 consecutive patients were listed for thyroid surgery at our surgical department. The study cohort consisted of 178 patients, divided into two groups based on the execution of the Valsalva manoeuvre. There was no difference in the duration of surgery between groups. Group B had fewer reinterventions for bleeding. Group A had a significantly greater volume of drainage output than Group B. Cervical haematoma can compromise a patient’s life, so bleeding control is crucial. Our results show that using a simple and safe Valsalva manoeuvre can improve the postoperative course with a significant reduction in drainage output, but does not prevent the risk of reoperation for hemorrhage.

## 1. Introduction

Total thyroidectomy (TT) is one of the most performed operations in the world. In neck surgery, it is among the most important interventions, as it is complex and very common [1].

Despite innovative approaches in surgery and innovations in surgical instruments, bleeding after TT remains a rare event affecting early postoperative morbidity, occurring in 0.3% to 4.2% of cases [2]. Post-TT bleeding is not only due to the neck’s anterolateral region’s complex anatomy, which is richly vascularised, but also to the strategies adopted to achieve perfect haemostasis in an operative field where contiguous anatomical structures must be preserved [3,4].

Intraoperative bleeding is an unpleasant complication that is usually easily manageable and resolvable, although it engages the surgeon and may lengthen operating time. In contrast, postoperative bleeding can represent a life-threatening condition for the patient; therefore, all known methods and strategies need to be used to minimise postoperative bleeding. Over time, the use of new instruments, the collagen-fibrinogen-thrombin patch (CFTP), cellulose gauze, and other hemostatic agents have made it possible to reduce the incidence of postoperative bleeding and reinterventions; however, it cannot be completely avoided [5,6].

The purpose of our study was to clarify whether the routine intraoperative execution of a Valsalva manoeuvre (VM) affected the detection of bleeding that would otherwise remain occult and possibly manifest in the postoperative period.

## 2. Methods

Between January 2019 to February 2022, 250 consecutive patients were listed for thyroid surgery at our surgical department. To evaluate the role of the VM, all patients treated using minimally invasive video-assisted thyroidectomy (MIVAT) or robot-assisted transaxillary thyroid surgery (RATS) were excluded from the study. Other exclusion criteria included thyroid lobectomies, the need for lymph node dissections, patient coagulation disorders, and high anaesthesiologic risk (ASA 3). We chronologically divided our cohort of 178 patients into 2 groups. MV was not performed in Group A (*n* = 96 patients); MV was performed at the end of the TT in Group B (*n* = 82 patients). This study was conducted according to STROBE guidelines (Figure 1) [7].

The VM was performed after thyroid excision and first revision of hemostasis on the operative field. The VM was achieved by applying an incremental PEEP (positive end expiratory pressure) up to 30 cm H_2_O for 30 s, followed by a new shorter VM performed by setting the mechanical ventilator to manual mode through the APL (adjustable pressure-limiting) valve and using the bag of the manual breathing unit (MBU).

All patients were treated using the same surgical approach by a team with extensive experience in neck surgery; therefore, all procedures were standardised.

The two groups had comparable demographic and clinical-pathological data (Table 1). Post hoc power was calculated to evaluate sample size (73.9% α 0.05).

Hemostasis was achieved using monopolar or bipolar coagulators, and LigaSure ™ (LSJ Medtronic, Covidien product, Minneapolis, MN, USA) was used to achieve better sealing of vessels in both groups. In dangerous areas, vessels’ ligation and application of small surgical clips was preferred to avoid recurrent laryngeal nerve injury.

At the end of the operation (after a VM execution in Group B), a human fibrinogen and human thrombin (CFTP) patch was always applied, and two suction surgical drains were placed in the thyroid lodges. A CFTP was also applied in Group A.

After the surgery, and anaesthesiologic awakening, patients were kept for 2 h in the recovery room, with personnel assigned to check clinical features and drainage outputs. Sometimes a longer observation time was required. If drains filled rapidly, or symptoms such as dyspnoea, a feeling of neck compression and smoothing of the jugule appeared, then the patient quickly underwent new surgery for haemostasis revision.

Primary outcomes considered included drainage volume, reoperation rates in the first 6 h, and in the first 24 h. The duration of the operation and the occurrence of complications other than bleeding were also evaluated.

All procedures performed in studies involving human participants were in accordance with the ethical standards of the institutional and/or national research committee and with the 1964 Helsinki Declaration and its later amendments or comparable ethical standards.

Informed written consent was obtained from all individual participants included in the study.

The study was registered on Clinicaltrial with the following ID: NCT05474261.

### Statistical Analysis

Data were analysed using the Statistical Package for the Social Sciences (SPSS) 28.0 (IBM Corp., Armonk, NY, USA). T tests for numerical variables and chi-square tests for categorical variables were performed to accomplish statistical comparison between groups. Phi Coefficients were calculated to determine the association between dichotomous variables. Point-biserial correlations were calculated to correlate between dichotomous and continuous variables. A binary logistic regression model was employed to determine the role of VMs in influencing postoperative bleeding (drainage volume). A *p* value less than 0.05 was considered statistically significant.

## 3. Results

The cohort was composed of 178 patients, divided into two groups based on the use of the VM. No statistically significant differences in age, sex, BMI (Body Mass Index), or ASA score (American Society of Anesthesiologists) between the two groups (*p* value > 0.05 for all; Table 1) were found.

Surgical and postoperative data are shown and analysed in Table 2. No difference in the duration of surgery between the two groups was recorded. Notably, Group B expressed fewer cases of both early (<6 h) and late re-intervention within the 24 h following the operation. These results were not statistically significant. The average period of stay under observation in the recovery room was also slightly shorter in Group B, but was not statistically significant. The results show that Group A was characterised by a significantly greater volume of drainage output in comparison with Group B, and that Group A patients were more frequently kept in place even on the second postoperative day. All patients had double suction drains placed in the right and left thyroid lodges.

Correlation tests (phi coefficients between dichotomous variables and point-biserial correlations between dichotomous and continuous variables) were used to evaluate the real contribution of the VM during TT. The results showed no significant correlation with early or late reoperation or persistence of drainage beyond the first postoperative day. In contrast, correlation with the overall volume of drainage output was statistically significant (Table 3). Postoperative bleeding (drainage volume) was further investigated using binary logistic regression to analyse the role of the VM (Table 4).

Finally, we analysed the incidence of major complications during TT (postoperative hypoparathyroidism, recurrent laryngeal nerve palsy, and wound infection). The two groups were homogeneous; no significant differences between the groups were found. These complications were not related to the VM application (Table 5). All patients included in this study had a 30-day follow-up period after surgery. All cases of hypoparathyroidism and recurrent laryngeal nerve palsy resolved during follow-up; as a result, they were deemed as transient.

## 4. Discussion

Total thyroidectomy and neck lymph node dissections are among the most frequent surgery performed at the cervical level and have low rates of morbidity and mortality. Bleeding after thyroidectomy is a well-known complication, but thanks to improvements in haemostasis, it has become a rare occurrence. However, despite its rarity, if bleeding is not diagnosed early and correctly managed it can become a potentially life-threatening condition [8,9,10].

In fact, patients with postoperative cervical haematoma are required to undergo a new surgery and lengthier hospital monitoring [11]. Symptoms arising include respiratory distress, choking, dysphagia, and a feeling of neck constriction. In 72% of cases, postoperative haematoma occurred within 6 h after surgery, and in 89% of cases, within 12 h [12,13]. The risk of bleeding was affected not only by patients’ features, but also by the underlying thyroid disease and the haemostatic techniques employed.

In our unit, all patients undergoing TT are routinely subjected to the placement of a drain in the right thyroid lodge and the left thyroid lodge; these drains are subsequently placed on suction. Drains are removed on the first postoperative day or later in cases of abnormal drainage output.

Traditionally, the main purpose of drains is to prevent postoperative complications when evacuating postoperative haematomas or lymphatic fluids and to notify the surgeon as soon as possible. However, drains usage may be omitted as unnecessary in uncomplicated cases because the drained volume is frequently very low, or because haemostasis was adequate during surgery [14,15]. In light of improvements in haemostatic techniques and increasing surgical skills, the value of drainage in thyroid surgery may again be questioned; using energy devices and advanced hemostatic agents could reduce drain output, making the use of drainage tubes less useful [16,17,18,19,20]. However, we believe that life-threatening complications (such as postoperative bleeding, haematoma, compression of the airways, or suffocation) can be more promptly signaled by the presence of drains, and can guarantee the surgeon a timelier diagnosis and a less rapid onset of symptoms associated with haematoma for patients [21,22].

The VM is a common procedure for detecting bleeding points during thyroidectomy procedures. During a VM, increased intrathoracic and intra-abdominal pressures cause internal jugular vein distension and an increase in internal jugular venous pressure. Venous hypertension involving the large veins causes an increase in blood flow to the vessels of the thyroid lodge, forcing any bleeding, and its detection [23,24].

Tokaç et al. suggest that intraoperative application of the VM has no positive effects on postoperative hemorrhagic complication [25]. In 2020, Beyoglu recommended maintaining airway pressure at 50 cm H_2_O for 22.5 s to achieve a more reliable intraoperative detection of bleeding points in patients undergoing total thyroidectomies [26]. Ozdemir affirms that the VM helps to detect bleeding points after Trendelenburg positioning and, given his experience, argues that manual compression is less effective [27].

In our study, we considered how the VM could help surgeons practicing TT. To reduce postoperative bleeding at the end of the haemostatic phase, we performed double VMs by applying an incremental PEEP up to 30 cm H_2_O for 30 s, followed by a new VM implemented by setting the mechanical ventilator to manual mode. The second shot was shorter and more energetic, and often identified a new bleeding point, allowing the surgeons to reach a clean and dry surgical field. All bleeding points found after the VMs were treated using monopolar or bipolar coagulators. Ties or stitching were preferred near the RLN. Revision of haemostasis after the VM was performed in 28 cases, or 34.15% of Group B cases. The average time required for re-hemostasis was negligible; it did not affect the overall operating time in our study.

The results reported herein should be considered in light of some limitations. Lymph node dissections increase the risk of postoperative bleeding [28]. The 178 patients enlisted for this study were suffering from benign and malignant pathologies. Forty-seven patients, accounting for 26% of the study sample, were treated for suspected malignant disease, and were therefore eligible for eventual lymph node dissection. The choice to execute total thyroidectomies without lymph nodal dissection was determined by the information gathered from patients’ clinical histories, US imaging features, the substantial uncertainty of malignant disease deriving from preoperative cytologic investigations. Further studies, including all thyroidectomies with lymph node dissections in the sample, are needed. The sample size was not sufficiently elevated, so the statistical power is less than 80%; it was not a double-blind study, so cultural and/or personal bias may have occurred. However, this procedure allowed us to have fewer re-operations both in the first 6 postoperative hours and in the following 24 h. Notably, many of the early reoperations (<6 h) were performed before the patient left the operating room or recovery room. The numerical increase in the reoperation rate in Group A might be explained by the exiguity of the sample and the patients’ selection criteria, the involvement of different surgeons despite their common experience, and/or rapidly filling surgical drains, any of which may have influenced the choice to re-operate.

These data were not statistically significant in our study, whereas a significant difference was recorded when quantifying the reduction in drainage volume and the lower incidence of drains’ permanence beyond the first postoperative day. The VM did not affect any of the other complications examined.

## 5. Conclusions

Haematoma manifestation may be life threatening, causing a severe compression of the airways; therefore, assertive bleeding control is crucial. Technological progress and using new haemostatic agents have contributed to a reduction in the incidence of this phenomenon. However, using a simple, economical, and safe manoeuvre such as the VM can improve the postoperative course with a significant reduction in drainage output, although it does not prevent the risk of reoperation.

## Figures and Tables

**Figure 1 jcm-11-05791-f001:**
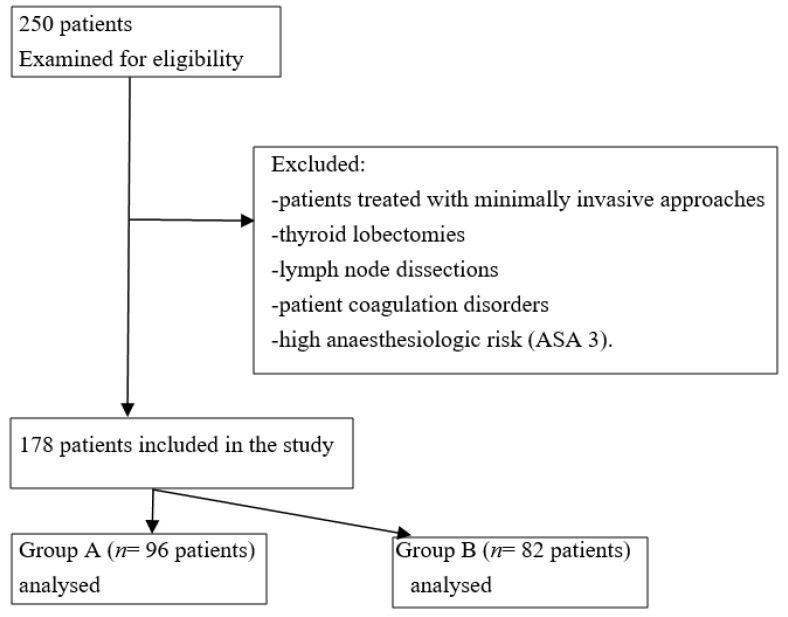
STROBE flowchart.

**Table 1 jcm-11-05791-t001:** Demographic data. BMI (Body Mass Index) and ASA score (American Society of Anesthesiologists). Significance tests include T tests for numerical variables and chi-square tests for categorical variables. A *p* value less than 0.05 was considered statistically significant.

	Group A (*n* = 96)	Group B (*n* = 82)	Overall (*n* = 178)	*p* Value
Age (Range)	20–74	18–74	18–74	
Mean	47.82	45.51	46.75	0.143
Median (±SD)	49.5 (±14.49)	45.5 (±14.28)	48 (±14.36)	
M/F	37/59	28/54	65/113	0.543
BMI (Range)	19–40	20–41	19–41	
Mean	25.23	25.25	25.24	0.501
Median (±SD)	26 (±4.08)	22 (±4.99)	25 (±4.55)	
ASA score				
I	38/96	35/82	63/178	0.675
II	58/96	47/82	115/178	
Multinodular Goiter	60/96	49/82	109/178	0.708
Grave’s disease	11/96	11/82	22/178	0.692
Malignant	25/96	22/82	47/178	0.905
Hashimoto’s thyroiditis	6/96	5/82	11/178	0.966

**Table 2 jcm-11-05791-t002:** Surgical and post-surgical data. The chi-square significance test was used. A *p* value less than 0.05 was considered statistically significant.

	Group A (*n* = 96)	Group B (*n* = 82)	*p* Value
Operation length (min)	126.11	126.78	0.416
Recovery room stay (min)	126.93	123.96	0.083
Reoperation (<6 h)	6/96	1/82	0.085
Reoperation (<24 h)	4/96	0/82	0.125
Drainage volume (mL)	79.38	56.67	<0.001
Permanence of drainages beyond the first postoperative day	16/96	5/82	0.029
Hospital stay (days)	2.22	2.15	0.175

**Table 3 jcm-11-05791-t003:** Correlation tests. Phi coefficients were calculated for associations between dichotomous variables. Point-biserial correlations were calculated for correlations between dichotomous and continuous variables. A *p* value less than 0.05 was considered statistically significant.

	Φ Value	*p* Value
Reoperation (<6 h)	−0.129	0.914
Reoperation (<24 h)	−0.140	0.938
Permanence of drainages on the second day	−0.163	0.970
	*r* Value	*p* Value
Drainage volume (mL)	−0.851	<0.001

**Table 4 jcm-11-05791-t004:** Binary logistic regression.

	B	S.E.	Exp (B)	95% C.I. EXP(B)		*p* Value
				Lower	Upper	
Drainage Volume	−0.462	0.136	0.630	0.482	0.822	<0.01
Constant	11.803	2.909	133,703.033			<0.01

**Table 5 jcm-11-05791-t005:** Most important complications during TT.

	Group A (*n* = 96)	Group B (*n* = 82)	*p* Value
Wound infection	2/96	1/82	0.655
Transient postoperative Hypoparathyroidism, *n* (%)	14/96	9/82	0.474
Transient recurrent laryngeal nerve Palsy, *n* (%)	2/96	2/82	0.873

## Data Availability

All data generated or analysed during this study are included in this article. Further enquiries can be directed to the corresponding author.

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
