# Peer review of "Clinical Usefulness of the Valsalva Manoeuvre to Improve Hemostasis during Thyroidectomy"

_jcm, 2022, doi:10.3390/jcm11195791_

Round 1

Reviewer 1 Report

The original article entitled “Clinical usefulness of the Valsalva Maneuver to improve hemostasis during thyroidectomy” analyzed 178 patients received total thyroidectomy and divided them in two groups based on the execution of the Valsalva Maneuver (VM), in order to clarify the role of VM and postoperative bleeding.

There are some major problems of this article:

1. The authors indicates that these 178 patients were all received total thyroidectomy (TT) in Line 66. However, receiving loboisthmectomy is one of exclusion criteria, which means there are a high proportion of TT in current group, and the authors should carefully describe the patient characteristics: benign/ malignant, with/without Graves disease or thyroiditis, with/without lymph node dissection (central/lateral).

2. In Table 1, the mean BMI of the patient population was over 30, the representativeness of the population used is questionable, and the effects of obesity on performing VM and postoperative bleeding rates is not described.

3. In Method section, the author did not well describe the detail of procedures.

(1) Are Ligasure small jaw and CFTP used routinely? If not, please stratified the data.

(2) The author should provide more detail about the execution of VMs. Please refer to Tokaç et al., Effect of Intraoperative Valsalva Maneuver Application on Bleeding Point Detection and Postoperative Drainage After Thyroidectomy Surgeries. Int Surg. 2015 Jun;100(6):994-8. (twice, 30 seconds of 30-cm PEEP)

(3) What is the definition of the need to re-hemostasis after intraoperative VM is performed? And the article does not mention the proportion of case need re-hemostasis and the time re-hemostasis needed.

4. In Table 2, the reoperation rate (6%) is too high in Group A (even if there were regular use of LSJ/CFTP), the author should carefully analyze this issue.

Author Response

Response to Reviewer 1 Comments

Dear reviewer,

first of all I want to thank you for having reviewed our work, which has engaged us in a long data collection, in a very important topic for our Surgical Unit.

Below are the responses to the comments you have kindly provided.

Glad to have worked with you, we look forward to any further clarifications.

Point 1 The authors indicates that these 178 patients were all received total thyroidectomy (TT) in Line 66. However, receiving loboisthmectomy is one of exclusion criteria, which means there are a high proportion of TT in current group, and the authors should carefully describe the patient characteristics: benign/ malignant, with/without Graves disease or thyroiditis, with/without lymph node dissection (central/lateral).

Response 1: Patient characteristics were added in Table 1 as required. Please note that the Track Changes function could not be used, but the added data was written in red. lymph node dissection (central/lateral) in an exclusion criteria.

Point 2: In Table 1, the mean BMI of the patient population was over 30, the representativeness of the population used is questionable, and the effects of obesity on performing VM and postoperative bleeding rates is not described.

Response 2: All of our data has been reviewed and the analyzes re-performed. The BMI results (Mean, Median, DS) were found to be erroneous due to the presence of filters in the table. We apologize for this mistake. The new data has been re-entered.

Point 3 : In Method section, the author did not well describe the detail of procedures.

(1) Are Ligasure small jaw and CFTP used routinely? If not, please stratified the data.

(2) The author should provide more detail about the execution of VMs. Please refer to Tokaç et al., Effect of Intraoperative Valsalva Maneuver Application on Bleeding Point Detection and Postoperative Drainage After Thyroidectomy Surgeries. Int Surg. 2015 Jun;100(6):994-8. (twice, 30 seconds of 30-cm PEEP)

(3) What is the definition of the need to re-hemostasis after intraoperative VM is performed? And the article does not mention the proportion of case need re-hemostasis and the time re-hemostasis needed.

Response 3: 1) Ligasure small jaw and CFTP are used routinely

2) More detail about the execution of VMs, in Method and Discussion Section

3) Details were added in the discussion section, as suggested.

Point 4  In Table 2, the reoperation rate (6%) is too high in Group A (even if there were regular use of LSJ/CFTP), the author should carefully analyze this issue.

Response 4  In our study we consider early reoperations (<6h), which often occur during the observation period in the recovery room (line 79-83). Sometimes a revision of the haemostasis was necessary even before the patient awakened from general anesthesia. We have addressed this issue, and we had an improvement of the data thanks to the application of the VM as reported in our study.

Reviewer 2 Report

I read with interest the article entitled “Clinical usefulness of the Valsalva Maneuver to improve hemostasis during thyroidectomy”by Mario Pacilli et all. because peri-operative bleeding after thyroid surgery constitutes an unresolved problem. In fact, its incidence results unchanged now for decades.

The article is well done, with sound statistical processing, consistent in its discussion and conclusions with respect to the results obtained.

Let me suggest some changes that can increase the quality of the paper.

-In Discussion, lines 131-132: Some references concerning the risks of thyroidectomy and lymph node dissection can be added in this regard:

*Delidze DD, and Coll:  A Narrative Review of Preventive Central Lymph Node Dissection in Patients With Papillary Thyroid Cancer - A Necessity or an Excess. Front Oncol. 2022, 12:906695. doi: 10.3389/fonc.2022.906695.

*Graceffa G, and Coll: Predictors of Central Compartment Involvement in Patients with Positive Lateral Cervical Lymph Nodes According to Clinical and/or Ultrasound Evaluation. J Clin Med, 2021 10(15):3407. doi: 10.3390/jcm10153407.

Moreover, surgery should be replaced with surgical procedures.

-In Discussion, line 142, you affirm “The use of drains in thyroid surgery is still widespread”. I do not completely agree with this statement, as the use of drains is very much dependent on the choices of individual teams, and several of them have discontinued their use for some time now, at least under standard conditions. Moreover, the use of energy devices and advanced hemostatic agents could reduce the drain output, thus making the use of drainage tubes less useful. 

This statement would therefore deserve more argumentation, supported by some references:

 *Tartaglia F, and coll:  Early discharge after total thyroidectomy: a retrospective feasibility study.

G Chir. 2016 ;37(6):250-256. doi: 10.11138/gchir/2016.37.6.250. 

*Soh TCF, Ong QJ, Yip HM: Complications of Neck Drains in Thyroidectomies: A Systematic Review and Meta-Analysis. Laryngoscope;131(3):690-700. doi: 10.1002/lary.29077.

*Docimo G, and Coll: A Gelatin-Thrombin Matrix Topical Hemostatic Agent (Floseal) in Combination With Harmonic Scalpel Is Effective in Patients Undergoing Total Thyroidectomy: A Prospective, Multicenter, Single-Blind, Randomized Controlled Trial. Surg. Innov, 2016, 23(1):23-9. doi: 10.1177/1553350615596638.  

*Canu GL, and Coll: The Use of Harmonic Focus and Thunderbeat Open Fine Jaw in Thyroid Surgery: Experience of a High-Volume Center. J Clin Med, 2022, 29;11(11):3062.  doi: 10.3390/jcm11113062.

*Scerrino G and Coll: Total thyroidectomy performed with the Starion vessel sealing system versus the conventional technique: a prospective randomized trial. Surg. Innov, 2010, 17(3):242-7. doi: 10.1177/1553350610376394.

-English is generally correct, however some errors or language flaws are found throughout the manuscript:

-Line 43-44 “Conseguently….bleeding”: this period is unclear and does not seem finalized. Please rephrase and replace "necessity" with "need".

-Line 55: It seems to sound better “thyroid lobectomy” instead of loboistmectomy.

-Lines 74-84: verbs concerning all actions should be conjugated in the past tense. Example: "drains were placed in the thyroid lodges...." etc.

-Line 90: Statisticalanalysis

-Line 99: replaceemploymentwith use

Author Response

Response to Reviewer 2 Comments

Dear reviewer,

first of all I want to thank you for having reviewed our work, which has engaged us in a long data collection, in a very important topic for our Surgical Unit.

Below are the responses to the comments you have kindly provided.

Glad to have worked with you, we look forward to any further clarifications.

Point 1 -In Discussion, lines 131-132: Some references concerning the risks of thyroidectomy and lymph node dissection can be added in this regard:

*Delidze DD, and Coll:  A Narrative Review of Preventive Central Lymph Node Dissection in Patients With Papillary Thyroid Cancer - A Necessity or an Excess. Front Oncol. 2022, 12:906695. doi: 10.3389/fonc.2022.906695.

*Graceffa G, and Coll: Predictors of Central Compartment Involvement in Patients with Positive Lateral Cervical Lymph Nodes According to Clinical and/or Ultrasound Evaluation. J Clin Med, 2021 10(15):3407. doi: 10.3390/jcm10153407.

Response 1: References are added, as required

Point 2 In Discussion, line 142, you affirm “The use of drains in thyroid surgery is still widespread”. I do not completely agree with this statement, as the use of drains is very much dependent on the choices of individual teams, and several of them have discontinued their use for some time now, at least under standard conditions. Moreover, the use of energy devices and advanced hemostatic agents could reduce the drain output, thus making the use of drainage tubes less useful.

This statement would therefore deserve more argumentation, supported by some references:

Response 2: Dear Reviewer,

the use of drains is habitual in our unit. We are aware that literature has dealt extensively with this issue, but ours is a choice of principle. We discussed the topic in the discussion section as required, and implemented the references.

Point 3

-Line 43-44 “Conseguently….bleeding”: this period is unclear and does not seem finalized. Please rephrase and replace "necessity" with "need".

-Line 55: It seems to sound better “thyroid lobectomy” instead of loboistmectomy.

-Lines 74-84: verbs concerning all actions should be conjugated in the past tense. Example: "drains were placed in the thyroid lodges...." etc.

-Line 90: Statisticalanalysis

-Line 99: replaceemploymentwith use

Response 3: All linguistic changes have been made, as suggested

Reviewer 3 Report

I have several comments as below:

1. The reoperation rate (6%) is too high in Group A, the author should explain this data, especially when Ligasure/CFTP were used.

2. In this study population, the proportion of patients undergoing total thyroidectomy was high, and the author needs to address how the patient selected. In addition, the authors should carefully describe patient characteristics, including pathology reports, thyroiditis, and lymph node dissections.

3. The authors should describe whether Ligasure small jaw and CFTP were used routinely, or a data stratification would be needed.

4. The authors should provide more detail about the procedure of Valsalva maneuver (VM) performed in this study in method section (how many times, how long the VM last).

5. The authors need to describe whether and when re-hemostasis is required after intraoperative VM, and also need to provide the proportion of cases requiring re-hemostasis and the time of re-hemostasis.

Author Response

Response to Reviewer 2 Comments

Dear reviewer,

first of all I want to thank you for having reviewed our work, which has engaged us in a long data collection, in a very important topic for our Surgical Unit.

Below are the responses to the comments you have kindly provided.

Glad to have worked with you, we look forward to any further clarifications.

Point 1 1. The reoperation rate (6%) is too high in Group A, the author should explain this data, especially when Ligasure/CFTP were used.

Response 1: In our study we consider early reoperations (<6h), which often occur during the observation period in the recovery room (line 79-83). Sometimes a revision of the haemostasis was necessary even before the patient awakened from general anesthesia. We have addressed this issue, and we had an improvement of the data thanks to the application of the VM as reported in our study.

Point 2. In this study population, the proportion of patients undergoing total thyroidectomy was high, and the author needs to address how the patient selected. In addition, the authors should carefully describe patient characteristics, including pathology reports, thyroiditis, and lymph node dissections.

Response 2: Patient characteristics were added in Table 1 as required. Please note that the Track Changes function could not be used, but the added data was written in red. lymph node dissection (central/lateral) in an exclusion criteria.

Point 3. The authors should describe whether Ligasure small jaw and CFTP were used routinely, or a data stratification would be needed.

Response 3: Ligasure small jaw and CFTP are used routinely

Point 4. The authors should provide more detail about the procedure of Valsalva maneuver (VM) performed in this study in method section (how many times, how long the VM last).

Response 4:  More detail about the execution of VMs, in Method and Discussion Section were added

Point 5. The authors need to describe whether and when re-hemostasis is required after intraoperative VM, and also need to provide the proportion of cases requiring re-hemostasis and the time of re-hemostasis.

Response 5:  Details were added in the discussion section, as suggested.

Reviewer 4 Report

This is a retrospective cohort study aimed at investigating the relationship between performing the Valsalva maneuver and occurrence of postoperative bleeding after total thyroidectomy.

The abstract is adequate, and has listed rationale and setting details, alongside the most important findings. I would recommend deleting the sentence on correlation tests, or naming the exact test. The conclusion does not flow from the results.

The objectives of the study are presented clearly and the introduction section communicates the need for investigating the impact of VM in thyroid surgery. Line 33 is unclear, as diffusion is a technical term describing something else. Line 45 – haemostatic is capitalized, and should not be.

I would suggest adding a STROBE flowchart and checklist, as required by ICMJE.

In addition, the cohort is of uneven size, and a power analysis is warranted to evaluate the sample size, since the incidence of bleeding is lower than the sample size.

A through language revision by a native speaker is recommended.

The data in the MM section warrant analysis by a binary logistic regression model, since the authors defined a binary outcome.

The limitations and shortcomings paragraph should also discuss the problems with sample sitze and patient selection bias.

The paper is well written, but suffers from a low sample size, and the conclusions do not flow directly from the results. A follow-up statistical analysis with a more comprehensive model is required.

Author Response

Response to Reviewer 3 Comments

Dear reviewer,

first of all I want to thank you for having reviewed our work, which has engaged us in a long data collection, in a very important topic for our Surgical Unit.

Below are the responses to the comments you have kindly provided.

Glad to have worked with you, we look forward to any further clarifications.

Point 1 The abstract is adequate, and has listed rationale and setting details, alongside the most important findings. I would recommend deleting the sentence on correlation tests, or naming the exact test. The conclusion does not flow from the results.

Response 1: The sentence is removed in the abstract

Point 2 The objectives of the study are presented clearly and the introduction section communicates the need for investigating the impact of VM in thyroid surgery. Line 33 is unclear, as diffusion is a technical term describing something else. Line 45 – haemostatic is capitalized, and should not be.

Response 2: The required corrections have been made.

Point 3 I would suggest adding a STROBE flowchart and checklist, as required by ICMJE.

In addition, the cohort is of uneven size, and a power analysis is warranted to evaluate the sample size, since the incidence of bleeding is lower than the sample size.

Response 3: STROBE flowchart and power analysis is added in method’s section.

Point 4 A through language revision by a native speaker is recommended.

Response 4:A linguistic revision by our translation and proofreading center was performed before submitting the paper

Point 5 The data in the MM section warrant analysis by a binary logistic regression model, since the authors defined a binary outcome.

Response 5: We are sorry but the statistical analysis cannot be implemented by our means. The tests used were compared with our statisticians. Point-Biserial Correlation test allows to measure the correlation between two variables in the circumstance that one of variables is dichotomous.

Point 6 The limitations and shortcomings paragraph should also discuss the problems with sample sitze and patient selection bias.

Response 6: limitations of the study are added and in Discussion Section.

Round 2

Reviewer 1 Report

English writing needed to be revised by native English speaker.

Author Response

Dear reviewer,

Our work has been reviewed by native English speakers. All linguistic adjustments, reported by the Editor, have been made, as requested.

Kind regards 

Reviewer 3 Report

The surgical quality is questionable. Using Ligasure/CFTP, the 6% reoperation rate in the non-Valsalva group is too high, which is not an issue of "reoperation" definition. The comparisons between two groups were not meaningful under a questionable surgical procedure, the significant difference may lie in a fundamental problem with surgical technique. 

There are also obvious problems with patient selection. After several exclusion, there were still 178 total thyroidectomy patients without lymph node dissection. The author did not well disclose the patient selection procedure in this article, and it is inappropriate. 

In addition, The authors still did not address the definition issues, including the implementation details of Valsalva maneuver, and the procedure about rehemostasis (for example, the indication, standard steps, and duration).

Author Response

Response to Reviewer 3 Comments

Dear reviewer,

first of all I want to thank you for having reviewed our work, which has engaged us in a long data collection, in a very important topic for our Surgical Unit.

Below are the responses to the comments you have kindly provided.

Glad to have worked with you, we look forward to any further clarifications.

Point 1: The surgical quality is questionable. Using Ligasure/CFTP, the 6% reoperation rate in the non-Valsalva group is too high, which is not an issue of "reoperation" definition. The comparisons between two groups were not meaningful under a questionable surgical procedure, the significant difference may lie in a fundamental problem with surgical technique. 

Response 2: This aspect is  commented discussion. It is actually true that the occurrence of hemorrhagic events in 6% of cases is higher than the average in the literature data: but our data are conditioned by limited group of patients, Selection and methodological  byas. The revised text is available In discussion section (line 199)

Point 2There are also obvious problems with patient selection. After several exclusion, there were still 178 total thyroidectomy patients without lymph node dissection. The author did not well disclose the patient selection procedure in this article, and it is inappropriate. 

Response 2: Total thyroidectomy is a specific choice in principle. Considering the whole cohort, only 26% were operated for a suspected or preoperatively confirmed malignant disease, and therefore deserving a lymph node dissection. More informations are added in the text as required(line 194).

Point 3: In addition, The authors still did not address the definition issues, including the implementation details of Valsalva maneuver, and the procedure about rehemostasis (for example, the indication, standard steps, and duration).

Response 3: VM is performed twice, the first through a gradual increase in PEEP that lasts 30 sec. The second time is a single shot by putting the mechanical ventilator in manual mode (APL valve), and using the bag of the manual breathing unit (MBU). More details have been added in the Method section (line 63-66) and in the Discussion Section (line 185-189).The rehemostasis phase, was implemented every time a new bleeding point arose (28/82, 34.15% of cases), step and duration are reported in discussion section (190-193).

Reviewer 4 Report

The authors have made some changes, improving the quality of the manuscript. Adding a power calculation and rephrasing the conclusions and the MM section has improved manuscript flow. 

However, the study still has a methodological issue - the Point Biserial test is a correlation measure, not a causality measure. The best way to conclude on something causing an outcome (such as bleeding) needs a probabilistic model, such as binary logistic regression. 

I would be keen on seeing the results presented that way.

Author Response

Response to Reviewer 4 Comments

Dear reviewer,

Point 1 - However, the study still has a methodological issue - the Point Biserial test is a correlation measure, not a causality measure. The best way to conclude on something causing an outcome (such as bleeding) needs a probabilistic model, such as binary logistic regression. 

I would be keen on seeing the results presented that way.

Response 1: The results have been updated, with the required statistical tests, thanks to your suggestions.
